# Approximation Based Variance Reduction for Reparameterization Gradients

**Tomas Geffner**
College of Information and Computer Science
University of Massachusetts, Amherst
tgeffner@cs.umass.edu

**Justin Domke**
College of Information and Computer Science
University of Massachusetts, Amherst
domke@cs.umass.edu

## Abstract

Flexible variational distributions improve variational inference but are harder to optimize. In this work we present a control variate that is applicable for any reparameterizable distribution with known mean and covariance matrix, e.g. Gaussians with any covariance structure. The control variate is based on a quadratic approximation of the model, and its parameters are set using a double-descent scheme by minimizing the gradient estimator's variance. We empirically show that this control variate leads to large improvements in gradient variance and optimization convergence for inference with non-factorized variational distributions.

## 1 Introduction

This paper concerns estimating the gradient of $\mathbb{E}_{q_w(\mathsf{z})} f(\mathsf{z})$ with respect to $w$. This is a ubiquitous problem in machine learning, needed to perform stochastic optimization in variational inference (VI), reinforcement learning, and experimental design [20, 12, 30, 5]. A popular technique is the "reparameterization trick" [24, 14, 8]. Here, one defines a mapping $\mathcal{T}_w$ that transforms some base density $q_0$ into $q_w$. Then, the gradient is estimated by drawing $\epsilon \sim q_0$ and evaluating $\nabla_w f(\mathcal{T}_w(\epsilon))$.

In any application using stochastic gradients, variance is a concern. Several variance reduction methods exist, with control variates representing a popular alternative [22]. A control variate is a random variable with expectation zero, which can be added to an estimator to cancel noise and decrease variance. Previous work has shown that control variates can significantly reduce the variance of reparameterization gradients, and thereby improve optimization performance [6, 17, 27].

Miller et al. [17] proposed a Taylor-expansion based control variate for the case where $q_w$ is a fully-factorized Gaussian parameterized by its mean and scale. Their method works well for the gradient with respect to the mean parameters. However, for the scale parameters, computational issues force the use of further approximations. In a new analysis (Sec. 4) we observe that this amounts to using a *constant* Taylor approximation (i.e. an approximation of order zero). As a consequence, for the scale parameters, the control variate has little effect. Still, the approach is very helpful with fully-factorized Gaussians, because in this case most variance is contributed by gradient with respect to the mean parameters.

The situation is different for non fully-factorized distributions: Often, most of the variance is contributed by the gradient with respect to the *scale* parameters. This renders Taylor-based control variates practically useless. Indeed, empirical results in Section 5 show that, with diagonal plus low rank Gaussians or Gaussians with arbitrary dense covariances, Taylor-based control variates yield almost no benefit over the no control variates baseline. (We generalize the Taylor approach to full-rank and diagonal plus low-rank distributions in Appendix E.3.)

For VI, fully factorized variational distributions are typically much less accurate than those representing interdependence [21, 32]. Thus, we seek a control variate that can aid the use of more powerful

distributions, such as Gaussians with any covariance structure (full-rank, factorized as diagonal plus low rank [21], Householder flows [32]), Student-t, and location-scale and elliptical families. This paper introduces such a control variate.

Our proposed method can be described in two steps. First, given any quadratic function $\hat{f}$ that approximates $f$, we define the control variate as $\mathbb{E}[\nabla_w \hat{f}(\mathcal{T}_w(\epsilon))] - \nabla_w \hat{f}(\mathcal{T}_w(\epsilon))$. We show that this control variate is tractable for any distribution with known mean and covariance. Intuitively, the more accurately $\hat{f}$ approximates $f$, the more this will decrease the variance of the original reparameterization estimator $\nabla_w f(\mathcal{T}_w(\epsilon))$. Second, we fit the parameters of $\hat{f}$ through a "double descent" procedure aimed at reducing the estimator's variance.

We empirically show that the use of our control variate leads to reductions in variance several orders of magnitude larger than the state of the art method when diagonal plus low rank or full-rank Gaussians are used as variational distributions. Optimization speed and reliability is greatly improved as a consequence.

## 2 Preliminaries

**Stochastic Gradient Variational Inference (SGVI).** Take a model $p(x, z)$, where $x$ is observed data and $z$ latent variables. The posterior $p(z|x)$ is often intractable. VI finds the parameters $w$ to approximate the target $p(z|x)$ with the simpler distribution $q_w(z)$ [12, 10, 2, 35]. It does this by maximizing the "evidence lower bound"

$$\text{ELBO}(w) = E_{q_w(\mathsf{z})} \log \frac{p(x, \mathsf{z})}{q_w(\mathsf{z})}, \tag{1}$$

which is equivalent to minimizing the KL divergence from the approximating distribution $q_w(z)$ to the posterior $p(z|x)$. Using $f(z) = \log p(x, z)$ and letting $\mathcal{H}(w) = -\mathbb{E}_{q_w(z)} \log q_w(z)$ denote the entropy, we can express the ELBO's gradient as

$$\nabla_w \text{ELBO}(w) = \nabla_w \mathbb{E}_{q_w(\mathsf{z})} f(\mathsf{z}) + \nabla_w \mathcal{H}(w). \tag{2}$$

SGVI's idea is that, while the first term from Eq. 2 typically has no closed-form, there are many unbiased estimators that can be used with stochastic optimization algorithms to maximize the ELBO [18, 23, 25, 31, 19, 27, 28, 29]. (We assume that the entropy term can be computed in closed form. If it cannot, one can "absorb" $\log q_w$ into $f$ and estimate its gradient alongside $f$.) These gradient estimators are usually based on the score function method [34] or the reparameterization trick [14, 31, 26]. Since the latter usually provides lower-variance gradients in practice, it is the method of choice whenever applicable. It requires a fixed distribution $q_0(\epsilon)$, and a transformation $\mathcal{T}_w(\epsilon)$ such that if $\epsilon \sim q_0(\epsilon)$, then $\mathcal{T}_w(\epsilon) \sim q_w(z)$. Then, an unbiased estimator for the first term in Eq. 2 is given by drawing $\epsilon \sim q_0(\epsilon)$ and evaluating

$$g(w, \epsilon) = \nabla_w f(\mathcal{T}_w(\epsilon)). \tag{3}$$

**Control Variates.** A control variate is a zero-mean random variable used to reduce the variance of another random variable [22]. Control variates are widely used in SGVI to reduce a gradient estimator's variance [17, 18, 33, 9, 25, 6, 3]. Let $g(w, \epsilon)$ define the base gradient estimator, using random variables $\epsilon$, and let the function $c(w, \epsilon)$ define the control variate, whose expectation over $\epsilon$ is zero. Then, for any scalar $\gamma$ we can get an unbiased gradient estimator as

$$g_{\text{cv}}(w, \epsilon) = g(w, \epsilon) + \gamma c(w, \epsilon). \tag{4}$$

The hope is that $c$ approximates and cancels the error in the gradient estimator $g$. It can be shown that the optimal weight is[1] $\gamma = -\mathbb{C}[c, g]/\mathbb{V}[c]$, which results in a variance of $\mathbb{V}[g_{\text{cv}}] = \mathbb{V}[g]\left(1 - \text{Corr}[c, g]^2\right)$. Thus, a good control variate will have high correlation with the gradient estimator (while still being zero mean). In the extreme case that $c = \mathbb{E}[g] - g$, variance would be reduced to zero. In practice, $\gamma$ must be estimated. This can be done approximately using empirical estimates of $\mathbb{E}[c^\top g]$ and $\mathbb{E}[c^\top c]$ from recent evaluations [6].

# 3 New Control Variate

This section presents our control variate. The goal is to estimate the gradient $\nabla_w \mathbb{E}_{q_w(z)} f(z)$ with low variance. The core idea behind our method is simple: if $f$ is replaced with a simpler function $\hat{f}$, a closed-form for $\nabla_w \mathbb{E}_{q_w(z)} \hat{f}(z)$ may be available. Then, the control variate is defined as the difference between the term $\nabla_w \mathbb{E}_{q_w(z)} \hat{f}(z)$ computed exactly and estimated using reparameterization. Intuitively, if the approximation $\hat{f}$ is good, this control variate will yield large reductions in variance.

We use a quadratic function $\hat{f}$ as our approximation (Sec. 3.1). The resulting control variate is tractable as long as the mean and covariance of $q_w$ are known (Sec. 3.2). While this is valid for any quadratic function $\hat{f}$, the effectiveness of the control variate depends on the approximation's quality. We propose to find the parameters of $\hat{f}$ by minimizing the final gradient estimator's variance $\mathbb{V}[g+c]$ or a proxy to it (Sec. 3.3). We do this via a double-descent scheme to simultaneously optimize the parameters of $\hat{f}$ alongside the parameters of $q_w$ (Sec. 3.4).

## 3.1 Definition, Validity, and Motivation

Given a function $\hat{f}_v$ that approximates $f$, we define the control variate as

$$c_v(w, \epsilon) = \nabla_w \mathbb{E}_{q_w(z)} \left[ \hat{f}_v(z) \right] - \nabla_w \hat{f}_v(\mathcal{T}_w(\epsilon)). \tag{5}$$

Since the second term is an unbiased estimator of the first one, $c_v(w, \epsilon)$ has expectation zero and thus represents a valid control variate. To understand the motivation behind this control variate consider the final gradient estimator,

$$g_{\text{cv}}(w, \epsilon) = g(w, \epsilon) + \gamma c_v(w, \epsilon) = \underbrace{\gamma \nabla_w \mathbb{E}_{q_w(z)} \left[ \hat{f}_v(z) \right]}_{\text{deterministic term}} + \underbrace{\nabla_w \left( f(\mathcal{T}_w(\epsilon)) - \gamma \hat{f}_v(\mathcal{T}_w(\epsilon)) \right)}_{\text{stochastic term}}. \tag{6}$$

Intuitively, making $\hat{f}_v$ a better approximation of $f$ will tend to make the stochastic term smaller, thus reducing the estimator's variance. We propose to set the approximating function to be a quadratic parameterized by $v$ and $z_0$,

$$\hat{f}_v(z) = b_v^\top (z - z_0) + \frac{1}{2}(z - z_0)^\top B_v (z - z_0), \tag{7}$$

where $b_v$ and $B_v$ are a vector and a square matrix parameterized by $v$, and $z_0$ is a vector. (We avoid including an additive constant in the quadratic since it would not affect the gradient.)

## 3.2 Tractability of the Control Variate

We now consider computational issues associated with the control variate from Eq. 5. Our first result is that, given $b_v$, $B_v$ and $z_0$, the control variate is tractable for any distribution with known mean and covariance. We begin by giving a closed-form for the expectation in Eq. 5 (proven in Appendix D).

**Lemma 3.1.** *Let $\hat{f}_v$ be defined as in Eq. 7. If $q_w$ has mean $\mu_w$ and covariance $\Sigma_w$, then*

$$\mathbb{E}_{q_w(z)} \hat{f}_v(z) = b_v^\top (\mu_w - z_0) + \frac{1}{2}\text{tr}(B_v \Sigma_w) + \frac{1}{2}\left( \mu_w^\top B_v \mu_w - z_0^\top B_v \mu_w - \mu_w^\top B_v z_0 + z_0^\top B_v z_0 \right). \tag{8}$$

If we substitute this result into Eq. 5, we can easily use automatic differentiation tools to compute the gradient with respect to $w$, and thus compute the control variate. Therefore, our control variate can be easily used for any reparameterizable distribution $q_w$ with known mean and covariance matrix. These include fully-factorized Gaussians, Gaussians with arbitrary full-rank covariance, Gaussians with structured covariance (e.g. diagonal plus low rank [21], Householder flows [32]), Student-t distributions, and, more generally, distributions in a location scale family or elliptical family.

**Computational cost.** The cost of computing the control variate depends on the cost of computing matrix-vector products (with matrix $B_v$) and the trace of $B_v \Sigma_w$ (see Eqs. 7 and 8). These costs depend on the structure of $B_v$ and $\Sigma_w$. We consider the case where $B_v$ and $\Sigma_w$ are parameterized as

diagonal plus low rank matrices, with ranks $r_v$ and $r_w$, respectively. Then, computing the control variate has cost $\mathcal{O}(d\,(1+r_v)\,(1+r_w))$, where $d$ is the dimensionality of $z$.

Notice that the cost of evaluating the reparameterization estimator $g(w,\epsilon)$ is at least $\mathcal{O}(d(1+r_w))$, since $\Sigma_w$ has $d(1+r_w)$ parameters. However, constant factors here are usually significantly higher than for the control variate, since evaluating $f$ requries a pass through a dataset. Thus, as long as $r_v$ is "small", the control variate does not affect the algorithm's overall scalability.

These complexity results extend to cases where $B_v$ and/or $\Sigma_w$ are diagonal or full-rank matrices by replacing the corresponding rank, $r_v$ or $r_w$, by 0 or $d$. For example, if $\Sigma_w$ is a full-rank matrix and $B_v$ is a diagonal plus rank-$r_v$, the control variate's cost is $\mathcal{O}(d^2 r_v)$. If both matrices are full-rank and $\Sigma_w$ is parameterized by its Cholesky factor $L$, the cost is $\mathcal{O}(d^3)$. This cubic cost comes entirely from instantiating $\Sigma_w = LL^\top$, all other costs are $\mathcal{O}(d^2)$.

### 3.3 Constructing the Quadratic Approximation

The results in the previous section hold for any quadratic function $\hat{f}_v$. However, for the control variate to reduce variance, it is important that $\hat{f}_v$ is a good approximation of $f$. This section proposes two methods to find such an approximation.

A natural idea would be to use a Taylor approximation of $f$ [17, 23]. However, as we discuss in Section 4, this leads to serious computational challenges (and is suboptimal). Instead, we will directly seek parameters $v$ that minimize the variance of the final gradient estimator $g_{\mathrm{cv}}$. For a given set of parameters $w$, we set $z_0 = \mu_w$ and find the parameters $v$ by minimizing an objective $\mathcal{L}_w(v)$. We present two different objectives that can be used:

**Method 1.** Find $v$ by minimizing the variance of the final gradient estimator (assuming $\gamma = 1$),

$$\mathcal{L}_w(v) = \mathbb{V}[g(w,\epsilon) + c_v(w,\epsilon)]. \tag{9}$$

Using a sample $\epsilon \sim q_0(\epsilon)$ an unbiased estimate of $\nabla_v \mathcal{L}_w(v)$ can be obtained as

$$h_w(\epsilon, v) = \nabla_v \|g(w,\epsilon) + c_v(w,\epsilon)\|^2. \tag{10}$$

**Method 2.** While the above method works well, it imposes a modest constant factor overhead, due to the need to differentiate through the control variate. As an alternative, we propose a simple proxy. The motivation is that the difference between the base gradient estimator and its approximation based on $\hat{f}_v$ is given by

$$\nabla_w f(\mathcal{T}_w(\epsilon)) - \nabla_w \hat{f}_v(\mathcal{T}_w(\epsilon)) = \left(\frac{d\,\mathcal{T}_w(\epsilon)}{d\,w}\right)^\top \left(\nabla f(\mathcal{T}_w(\epsilon)) - \nabla \hat{f}_v(\mathcal{T}_w(\epsilon))\right). \tag{11}$$

Thus, the closer $\nabla \hat{f}_v(z)$ is to $\nabla f(z)$, the better the control variate $c_v$ can approximate and cancel estimator $g$'s noise. Accordingly, we propose the proxy objective

$$\mathcal{L}_w(v) = \frac{1}{2} \mathop{\mathbb{E}}_{q_0(\epsilon)} ||\nabla f(\mathcal{T}_w(\epsilon)) - \nabla \hat{f}_v(\mathcal{T}_w(\epsilon))||^2. \tag{12}$$

Using a sample $\epsilon \sim q_0(\epsilon)$ an unbiased estimate of $\nabla_v \mathcal{L}_w(v)$ can be obtained as

$$h_w(\epsilon, v) = \frac{1}{2}\nabla_v ||\nabla f(\mathcal{T}_w(\epsilon)) - \nabla \hat{f}_v(\mathcal{T}_w(\epsilon))||^2. \tag{13}$$

We observed that both methods lead to reductions in variance of similar magnitude (see Fig. 1 for a comparison). However, the second method introduces a smaller overhead.

The idea of using a double-descent scheme to minimize gradient variance was explored in previous work. It has been done to set the parameters of a sampling distribution [29], and to set the parameters of a control variate for discrete latent variable models [9, 33] using a continuous relaxation for discrete distributions [11, 16].

### 3.4 Final Algorithm

This section presents an efficient algorithm to use our control variate for SGVI. The approach involves maximizing the ELBO and finding a good quadratic approximation $\hat{f}_v$ simultaneously, via a double-descent scheme. We maximize the ELBO using stochastic gradient ascent with the gradient estimator

**Algorithm 1** SGVI with the proposed control variate.

---
**Require:** Learning rates $\alpha^{(w)}, \alpha^{(v)}$.
   Initialize $w_0$, $v_0$ and control variate weight $\gamma = 0$.
   **for** $k = 1, 2, \cdots$ **do**
      Sample $\epsilon \sim q_0$ and compute $z = \mathcal{T}_{w_k}(\epsilon)$.
      Compute estimator and control variate $g = g(w_k, \epsilon)$, $c = c_{v_k}(w_k, \epsilon)$.           (Eqs. 3 and 5)
      Take primary step as $w_{k+1} \leftarrow w_k + \alpha^{(w)}(g + \gamma c)$.
      Update $\gamma$ to minimize empirical $\mathbb{V}[g + \gamma c]$.                    (Sec. 3.4)
      Compute control variate gradient estimator $h = h_w(\epsilon, v_k)$.           (Eq. 10 or 13)
      Take dual step as $v_{k+1} \leftarrow v_k - \alpha^{(v)}h$.
   **end for**

---

from Eq. 3 and our control variate for variance reduction. Simultaneously, we find an approximation $\hat{f}_v$ by minimizing $\mathcal{L}_w(v)$ using stochastic gradient descent with the gradient estimators from Eq. 10 or 13. Our procedure, summarized in Alg. 1, involves alternating steps of each optimization process. Notably, optimizing $v$ as in Alg. 1 does not involve extra likelihood evaluations, since the model evaluations used to estimate the ELBO's gradient are re-used to estimate $\nabla_v \mathcal{L}_w(v)$.

Alg. 1 includes the control variate weight $\gamma$. This is useful in practice, specially at the beginning of training, when $v$ is far from optimal and $\hat{f}_v$ is a poor approximation of $f$. The (approximate) optimal weight can be obtained by keeping estimates of $\mathbb{E}[c(w, \epsilon)^\top g(w, \epsilon)]$ and $\mathbb{E}[c(w, \epsilon)^\top c(w, \epsilon)]$ as optimization proceeds [6].

## 4 Comparison of Approximations

**Taylor-Based Approximations.** There is closely related work exploring Taylor-expansion based control variates for reparameterization gradients [17]. These control variates can be expressed as

$$c(w, \epsilon) = \underset{q_0(\epsilon)}{\mathbb{E}} \left[ \left( \frac{d\mathcal{T}_w(\epsilon)}{dw} \right)^\top \nabla \hat{f}(\mathcal{T}_w(\epsilon)) \right] - \left( \frac{d\mathcal{T}_w(\epsilon)}{dw} \right)^\top \nabla \hat{f}(\mathcal{T}_w(\epsilon)). \qquad (14)$$

This is similar to Eq. 5. The difference is that, here, the approximation $\hat{f}$ is set to be a Taylor expansion of $f$. In general this leads to an intractable control variate: the expectation may not be known, or the Taylor approximation may be intractable (e.g. requires computing Hessians). However, in some cases, it can be computed efficiently. For this discussion we focus on Gaussian variational distributions, where the parameters $w$ are the mean and scale.

For the gradient with respect to the mean parameters, $\hat{f}(z)$ can be set to be a second-order Taylor expansion of $f(z)$ around the current mean. This might appear to be problematic, since computing the Hessian of $f$ will be intractable in general. However, it turns out that, for the mean parameters, this leads to a control variate that can be computed using only Hessian-vector products. This was first observed by Miller et al. [17] for diagonal Gaussians.

For the scale parameters, even with a diagonal Gaussian, using a second-order Taylor expansion requires the diagonal of the Hessian, which is intractable in general. For this reason, Miller et al. [17] propose an approach equivalent[2] to setting $\hat{f}$ to a *first*-order Taylor expansion, so that $\nabla \hat{f}$ is constant.

The biggest drawback of Taylor-based control variates is that the crude first-order Taylor approximation used for the scale parameters provides almost no variance reduction. Interestingly, this seems to pose very little problem with diagonal Gaussians. This is because, in this case, the gradient with respect to the mean parameters typically contribute almost all the variance. However, this approach may be useless in some other situations: With non-diagonal distributions, the scale parameters often contribute the majority of the variance (see Fig. 1).

A second drawback is that even a second-order Taylor expansion is not optimal. A Taylor expansion provides a good *local* approximation, which may be poor for distributions $q_w$ with large variance.

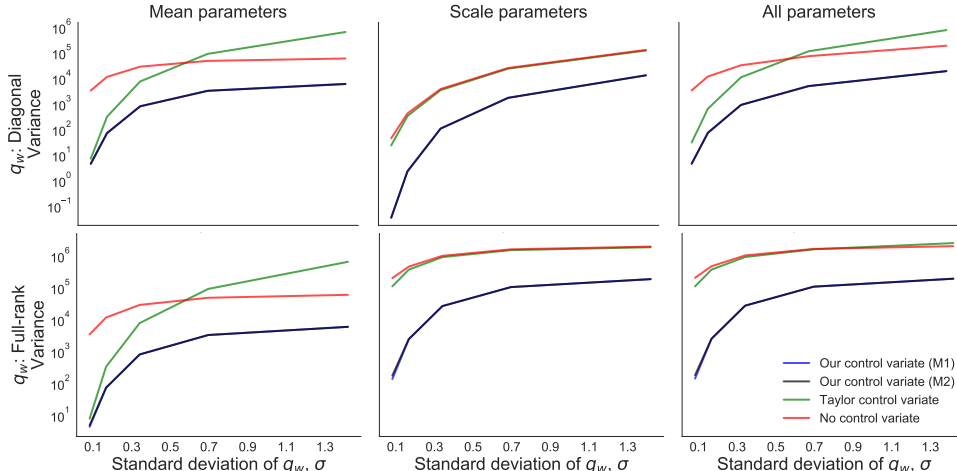

Figure 1: **The new control variate improves variance, particularly for scale parameters.** Variance of different gradient estimators on a Bayesian logistic regression model for a variational distribution with mean zero and covariance $\sigma^2 I$ for varying $\sigma$. The mean parameters (where Miller's approach often works well) dominate the variance for fully-factorized distributions, while the scale parameters (where Miller's approach does little) dominate for full-rank Gaussians. Method 1 (M1) and Method 2 (M2) to find the parameters of our control variate perform extremely similarly.

**Demonstration.** Fig. 1 compares four gradient estimators on a Bayesian logistic regression model (see Sec. 5): plain reparameterization, reparameterization with a Taylor-based control variate, and reparameterization with our control variate (minimizing Eq. 9 or Eq. 12, using a diagonal plus rank-10 matrix $B_v$). The variational distribution is either a diagonal Gaussian or a Gaussian with arbitrary full-rank covariance. We set the mean $\mu_w = 0$ and covariance $\Sigma_w = \sigma^2 I$. We measure each estimator's variance for different values of $\sigma$. For transparency, in all cases we use a fixed weight $\gamma = 1$.

There are four key observations: (i) Our variance reduction for the mean parameters is somewhat better than a Taylor approximation (which even increases variance in some cases). This is not surprising, since a Taylor expansion was never claimed to be optimal; (ii) Our control variate is vastly better for the scale parameters; (iii) the variance for fully-factorized distributions is dominated by the mean, while the variance for full-covariance distributions it is dominated by the scale; (iv) the proxy for the gradient variance (Eq. 9) performs extremely similarly to the true gradient variance (Eq. 12).

It should be emphasized that, for this analysis, the parameters $v$ are trained to completion for each value of $\sigma$. This does not exactly reflect what would be expected in practice, where the dual-descent scheme "tracks" the optimal $v$ as $w$ changes. Experiments in the next section consider this practical setting.

## 5 Experiments and Results

We present results that empirically validate the the proposed control variate and algorithm. We perform SGVI on several probabilistic models using different variational distributions. We maximize the ELBO using the reparameterization estimator with the proposed control variate to reduce its variance (Alg. 1). We compare against optimizing using the reparameterization estimator without any control variates, and against optimizing using a Taylor-based control variates for variance reduction.

### 5.1 Experimental details

**Tasks and datasets:** We use three different models: Logistic regression with the *a1a* dataset, hierarchical regression with the *frisk* dataset [7], and a Bayesian neural network with the *red wine* dataset. The latter two are the ones used by Miller et al. [17]. (Details for each model in App. C.)

| #Samples | Model | $q_w$: Diag plus low rank | | | $q_w$: full-rank covariance | | |
|---|---|---|---|---|---|---|---|
| | | Base | Our CV | Taylor | Base | Our CV | Taylor |
| $M = 10$ | Hierarchical | 4.4 | 6.4 | 10.8 | 3.9 | 6.0 | 10.2 |
| | Logistic | 3.8 | 6.3 | 7.7 | 4.9 | 8.1 | 9.7 |
| | BNN | 11.1 | 16.2 | 31.2 | – | – | – |
| $M = 50$ | Hierarchical | 5.8 | 8.3 | 12.8 | 4.9 | 7.4 | 11.7 |
| | Logistic | 8.1 | 11 | 16.5 | 14.2 | 20.1 | 32.1 |
| | BNN | 17.3 | 25.6 | 48.4 | – | – | – |

Table 1: Cost (milliseconds) of performing one optimization step using no control variates (Base), a Taylor-based control variate (Taylor), and our control variate (Our CV). For the latter, one step involves computing the gradient, control variate, and updating the parameters $v$. For reference, computing the Hessian of $f$ takes $131, 146$ and $2883$ milliseconds for the hierarchical regression, logistic regression and Bayesian neural network models. As expected, because of these high costs, using a second order Taylor-based control variate for the scale parameters is not practical.

**Variational distribution:** We consider diagonal Gaussians parameterized by the log-scale parameters, and diagonal plus rank-10 Gaussians, whose covariance is parameterized by a diagonal component $D$ and a factor $F$ of shape $d \times 10$ (i.e. $\Sigma_w = D + FF^\top$) [21]. For the simpler models, logistic regression and hierarchical regression, we also consider full-rank Gaussians parameterized by the Cholesky factor of the covariance.

**Algorithmic details:** We use Adam [13] to optimize the parameters $w$ of the variational distribution $q_w$ (with step sizes between $10^{-5}$ and $10^{-2}$). We use Adam with a step size of $0.01$ to optimize the parameters $v$ of the control variate, by minimizing the proxy to the variance from Eq. 12. We parameterize $B_v$ as a diagonal plus rank-$r_v$. We set $r_v = 10$ when diagonal or diagonal plus low rank variational distributions are used, and $r_v = 20$ when a full-rank variational distribution is used. (We show results using other ranks in Appendix B.)

**Baselines considered:** We compare against optimization using the base reparameterization estimator (Eq. 3). We also compare against using Taylor-based control variates. (We generalize the Taylor approach to full-rank and diagonal plus low-rank distributions in Appendix E.3.) For all control variates we find the (approximate) optimal weight using the method from Geffner and Domke [6] (fixing the weight to 1 lead to strictly worse results). We use $M = 10$ and $M = 50$ samples from $q_w$ to estimate gradients.

We show results in terms of iterations and wall-clock time. Table 1 shows the per iteration time-cost of each method in our experiments. Our method's overhead is around 50%, while the Taylor approach has an overhead of around 150%. These numbers depend on the implementation and platform, but should give a rough estimate of the overhead in practice (we use PyTorch 1.1.0 on an Intel i5 2.3GHz).

## 5.2 Results

Fig. 2 shows optimization results for the diagonal plus low rank Gaussian variational distribution. The two leftmost columns show ELBO vs. iteration plots for two specific learning rates. The third column shows, for each method and iteration, the ELBO for the best learning rate chosen retrospectively. In all cases, our method improves over competing approaches. In fact, our method with $M = 10$ samples to estimate the gradients performs better than competing approaches with $M = 50$. On the other hand, Taylor-based control variates give practically no improvement over using the base estimator alone. This is because most of the gradient variance comes from estimating the gradient with respect to the scale parameters, for which Taylor-based control variates do little.

To test the robustness of optimization, in Fig. 3 we show the final training ELBO after 80000 steps as a function of the step size used. Our method is less sensitive to the choice of step size. In particular, our method gives reasonable results with larger learning rates, which translates to better results with a smaller number of iterations.

For space reasons, results for diagonal Gaussians and Gaussians with arbitrary full-rank covariances as variational distributions are shown in Appendix A. Results for full-rank Gaussians are similar to

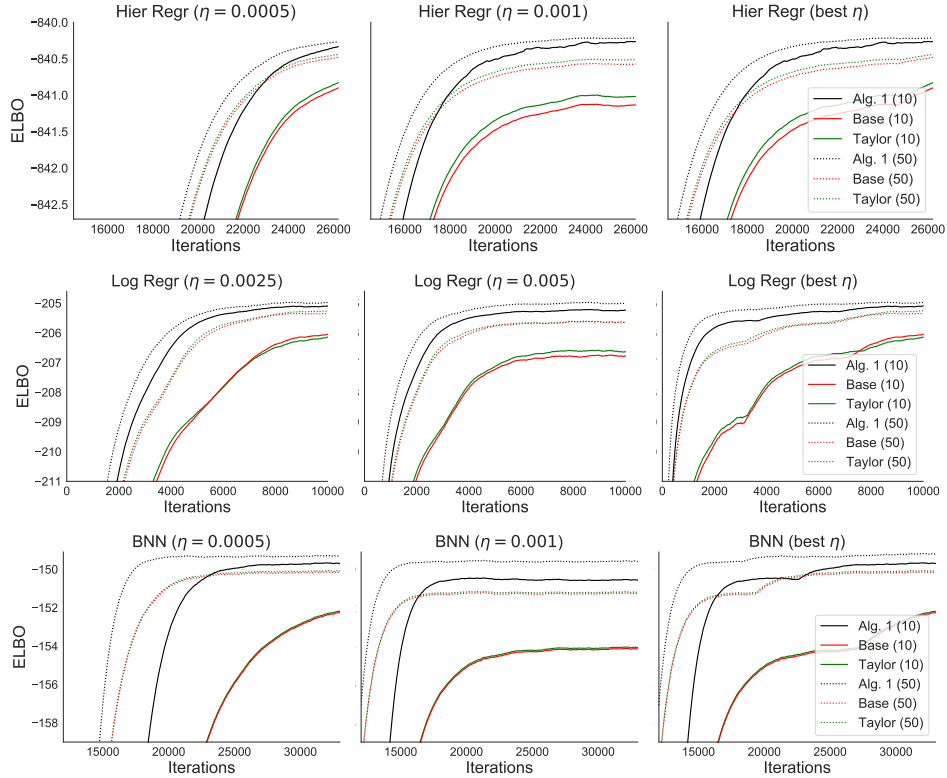

Figure 2: **The use of our control variate yields improved optimization convergence.** VI using a diagonal plus low rank Gaussian variational distribution. The first two columns show results for two different step-sizes, and the third one using the best step-size chosen retrospectively. "Base (M)" stands for the base reparameterization gradient estimated using $M$ samples, and "Taylor (M)" for using a Taylor-expansion based control variate for variance reduction.

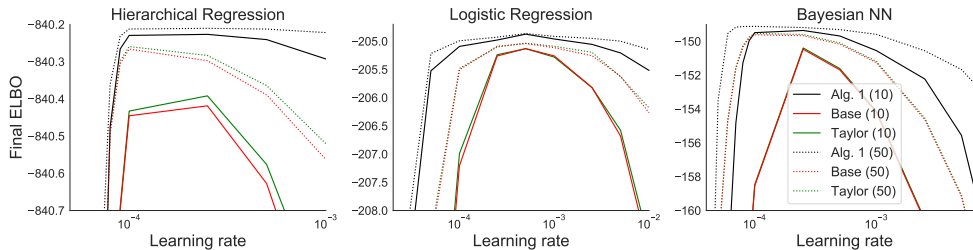

Figure 3: **The use of our control variate yields good results for a wider range of step sizes.** VI using a diagonal plus low rank covariance Gaussian variational distribution. The plots show the final ELBO achieved after training for 80000 steps vs. step size used. (Higher ELBO is better.)

the ones shown in Figs. 2 and 3. Our method performs considerably better than competing approaches (our method with $M = 10$ outperforms competing approaches with $M = 50$), and Taylor-based control variates yield no improvement over the no control variate baseline.

On the other hand, with diagonal Gaussians, our approach and Taylor-based control variates perform similarly – both are significantly better than the no control variate baseline. We attribute the success of Taylor-based approaches in this case to two related factors. First, diagonal approximations tend to under-estimate the true variance, so a local Taylor approximation may be more effective. Second, for diagonal Gaussians most of the gradient variance comes from mean parameters, where a second-order Taylor approach is tractable.

### 5.2.1 Estimator's Variance as Optimization Proceeds

Fig. 1 showed a comparison of the variance reduction achieved by our control variate in the ideal setting for which the control variate's parameters $v$ were fully optimized at every step. While insightful, the analysis did not reflect how the control variate is used in practice, where the parameters $v$ "track" the optimal parameters as $w$ changes via a double-descent scheme. We now show results for this practical setting. We set the variational distribution to be a diagonal plus low rank Gaussian and perform optimization with each of the gradient estimators. We estimate the variance of each estimator as optimization proceeds. Results are shown in Fig. 4. It can be observed that our control variate yields variance reductions of several orders of magnitude, while Taylor-based control variates lead to barely any variance reduction at all. This is aligned with our previous analysis and results.

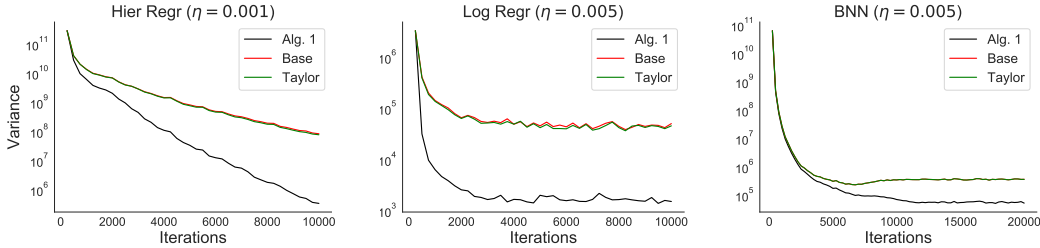

Figure 4: **The use of our control variate yields large reductions in variance.** Variance of different gradient estimators as optimization proceeds for the three models considered. "Base" stands for the base reparameterization gradient, and "Taylor" for using a Taylor-expansion based control variate for variance reduction. For the BNN model the lines for the base estimator and the Taylor control variate are almost completely overlapped, and thus indistinguishable in the plot. (All methods have the same variance at initialization because control variate weights are initialized to 0.)

### 5.2.2 Wall-clock Time Results

Fig. 5 shows results in terms of wall-clock time instead of iterations. These results' main purpose is visualization, they are the same as the ones in Fig. 2 (right column) with the x-axis scaled for each estimator with the values from Table 1.

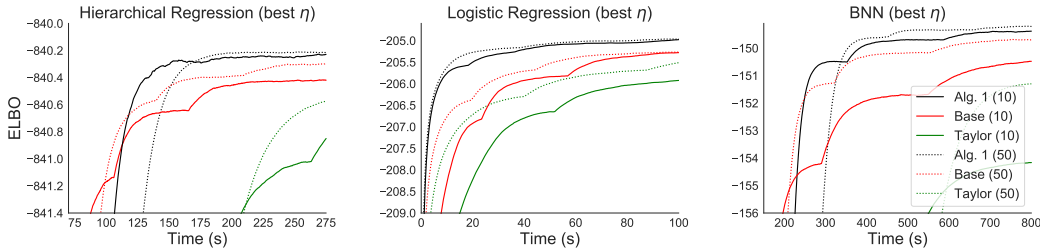

Figure 5: **The use of our control variate yields improved optimization convergence.** VI using a diagonal plus low rank Gaussian variational distribution, with the best step-size chosen retrospectively for each time horizon. "Base ($M$)" stands for the base reparameterization gradient estimated using $M$ samples, and "Taylor ($M$)" for using a Taylor-expansion based control variate for variance reduction.

Finally, it is worth mentioning that our control variate may be used jointly with other variance reduction methods. For instance, the sticking-the-landing (STL) estimator [27] can be used with our control variate in two ways: (i) setting the base gradient estimator to be the STL estimator; and (ii) creating the "STL control variate" and using in concert with our control variate [6]. In addition, while we focus on reparameterization, our control variate could be used with other estimators as well, such as the score function or generalized reparameterization [28], as long as the covariance of the variational distribution is known. This is done by obtaining the second term from eq. 5 using the corresponding estimator (instead of reparameterization).

## Broader Impact

In this work we present a new algorithm that yields improved performance for VI with non factorized distributions. We believe this algorithm could be included in VI-based automatic inference tools to improve their performance. This could have an impact in several areas since these tools, such as ADVI [15] (in Stan [4]), are used by researchers and practitioners in many different fields.

**Acknowledgments and Disclosure of Funding.** This material is based upon work supported by the National Science Foundation under Grant No. 1908577.

## Footnotes

[1]Since $g$ and $c$ are vectors, the expressions for $\gamma$ and $\mathbb{V} g_{\text{cv}}$ should be interpreted using $\mathbb{V} X = \mathbb{E} \|X\|^2 - \|\mathbb{E} X\|^2$, $\mathbb{C}[X, Y] = \mathbb{E}[(X - \mathbb{E} X)^\top (Y - \mathbb{E} Y)]$, and $\text{Corr}[X, Y] = \text{Cov}[X, Y]/\sqrt{\mathbb{V}[X]\mathbb{V}[Y]}$

[2]The original paper [17] describes the control variate for the scale parameters as using a second-order Taylor expansion, and then applies an additional approximation based on a minibatch to deal with intractable Hessian computations. In Appendix E we show these formulations are exactly equivalent.

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
