[Supplementary Material]

# A   Results with Other Variational Distributions

## A.1   Gaussian with Arbitrary Full-rank Covariance Variational Distribution

Figure 6: VI using a Gaussian with a full-rank covariance. The first two columns show results for two different step-sizes, and the third one using the best step-size chosen retrospectively. (Higher ELBO is better.)

Figure 7: VI using a Gaussian with a full-rank covariance. The plots show the final ELBO achieved after training for 80000 steps vs. step size used. (Higher ELBO is better.)

Figure 8: VI using a Gaussian with a full-rank covariance, with the best step-size chosen retrospectively. (Higher ELBO is better.)

## A.2 Fully-factorized Gaussian Variational Distribution

Figure 9: VI using a fully-factorized Gaussian. The first two columns show results for two different step-sizes, and the third one using the best step-size chosen retrospectively. (Higher ELBO is better.)

Figure 10: VI using a fully-factorized Gaussian. The plots show the final ELBO achieved after training for 40000 steps vs. step size used. (Higher ELBO is better.)

Figure 11: VI using a diagonal Gaussian, with the best step-size chosen retrospectively. (Higher ELBO is better.)

# B Results for Other Ranks

Fig. 12 shows results obtained using different values for the control variate's rank $r_v$. For clarity, in all cases we use $M = 10$ and we do not include results obtained using the Taylor expansion based control variate. It can be observed that the control variate leads to improved performance for a wide range of ranks. However, using a rank that is too low may hinder its benefits considerably (this can be clearly seen for the logistic regression model).

Figure 12: VI using a diagonal plus low rank Gaussian, using different ranks for our control variate.

# C Models Used

**Bayesian logistic regression:** We use a subset of $700$ rows of the *a1a* dataset. In this case the posterior $p(z|x)$ has dimensionality $d = 120$. Let $\{x_i, y_i\}$, where $y_i$ is binary, represent the *i*-th sample in the dataset. The model is given by

$$
\begin{aligned}
w_i &\sim \mathcal{N}(0, 1), \\
p_i &= (1 + \exp(w_0 + w \cdot x_i))^{-1}, \\
y_i &\sim \text{Bernoulli}(p_i).
\end{aligned}
$$

**Hierarchical Poisson model:** By Gelman et al. [7]. The model measures the relative stop-and-frisk events in different precincts in New York city, for different ethnicities. In this case the posterior $p(z|x)$ has dimensionality $d = 37$. The model is given by

$$
\begin{aligned}
\mu &\sim \mathcal{N}(0, 10^2) \\
\log \sigma_\alpha &\sim \mathcal{N}(0, 10^2), \\
\log \sigma_\beta &\sim \mathcal{N}(0, 10^2), \\
\alpha_e &\sim \mathcal{N}(0, \sigma_\alpha^2), \\
\beta_p &\sim \mathcal{N}(0, \sigma_\beta^2), \\
\lambda_{ep} &= \exp(\mu + \alpha_e + \beta_p + \log N_{ep}), \\
Y_{ep} &\sim \text{Poisson}(\lambda_{ep}).
\end{aligned}
$$

Here, $e$ stands for ethnicity, $p$ for precinct, $Y_{ep}$ for the number of stops in precinct $p$ within ethnicity group $e$ (observed), and $N_{ep}$ for the total number of arrests in precinct $p$ within ethnicity group $e$ (which is observed).

**Bayesian neural network:** As done by Miller et al. [17] we use a subset of 100 rows from the "Red-wine" dataset. We implement a neural network with one hidden layer with 50 units and Relu activations. In this case the posterior $p(z|x)$ has dimensionality $d = 653$. Let $\{x_i, y_i\}$, where $y_i$ is an integer between one and ten, represent the $i$-th sample in the dataset. The model is given by

$$
\begin{aligned}
\log \alpha &\sim \text{Gamma}(1, 0.1), \\
\log \tau &\sim \text{Gamma}(1, 0.1), \\
w_i &\sim \mathcal{N}(0, 1/\alpha), \qquad\qquad\quad \text{(weights and biases)} \\
\hat{y}_i &= \text{FeedForward}(x_i, W), \\
y_i &\sim \mathcal{N}(\hat{y}_i, 1/\tau).
\end{aligned}
$$

# D   Proof of Lemma

**Lemma D.1.** *Let $\hat{f}(z)$ be defined as in Eq. 7. If $q_w(z)$ is a distribution with mean $\mu_w$ and covariance matrix $\Sigma_w$, then*

$$
\mathbb{E}_{q_w(z)} \hat{f}_v(z) = b_v^\top(\mu_w - z_0) + \frac{1}{2}\text{tr}(B_v\Sigma_w) + \frac{1}{2}\left(\mu_w^\top B_v \mu_w - z_0^\top B_v \mu_w - \mu_w^\top B_v z_0 + z_0^\top B_v z_0\right) \quad (15)
$$

*Proof.* We have

$$
\hat{f}(z) = b^\top(z - z_0) + \frac{1}{2}(z - z_0)^\top B(z - z_0).
$$

Taking the expectation with respect to $q_w(z)$ gives

$$
\mathbb{E}_{q_w(z)} \hat{f}(z) = b^\top(\mu_w - z_0) + \frac{1}{2}\underbrace{\mathbb{E}_{q_w(z)}[(z - z_0)^\top B(z - z_0)]}_{t(w)} \quad (16)
$$

We now deal with the term in the second line of Eq. 16, $t(w)$.

$$
\begin{aligned}
t(w) &= \mathbb{E}_{q_w(z)}[(z - z_0)^\top B(z - z_0)] \\
&= \mathbb{E}_{q_w(z)}\left[\text{tr}\left((z - z_0)^\top B(z - z_0)\right)\right] \\
&= \mathbb{E}_{q_w(z)}\left[\text{tr}\left(B(z - z_0)(z - z_0)^\top\right)\right] \\
&= \text{tr}\left(B\,\mathbb{E}_{q_w(z)}[(z - z_0)(z - z_0)^\top]\right) \\
&= \text{tr}\left(B\,\mathbb{E}_{q_w(z)}[zz^\top - zz_0^\top - z_0 z^\top + z_0 z_0^\top]\right) \\
&= \text{tr}\left(B\,\mathbb{E}_{q_w(z)}[zz^\top - zz_0^\top - z_0 z^\top + z_0 z_0^\top]\right) \\
&= \text{tr}\left(B\,\mathbb{E}[(z - \mu_w + \mu_w)(z - \mu_w + \mu_w)^\top - zz_0^\top - z_0 z^\top + z_0 z_0^\top]\right) \\
&= \text{tr}\left(B\left(\mathbb{E}[(z - \mu_w)(z - \mu_w)^\top] + \mu_w\mu_w^\top - \mu_w z_0^\top - z_0\mu_w^\top + z_0 z_0^\top\right)\right) \\
&= \text{tr}\left(B\left(\Sigma_w + \mu_w\mu_w^\top - \mu_w z_0^\top - z_0\mu_w^\top + z_0 z_0^\top\right)\right) \\
&= \text{tr}(B\Sigma_w) + \mu_w^\top B\mu_w - z_0^\top B\mu_w - \mu_w^\top Bz_0 + z_0^\top Bz_0.
\end{aligned}
$$

Combining Eq. 16 with the expression for $t(w)$ completes the proof. $\qquad\qquad\square$

# E  Details on Taylor-based Control Variates

There is closely related work exploring Taylor-expansion based control variates for reparameterization gradients by Miller et al. [17]. They develop a control variate for the case where $q_w$ is a fully-factorized Gaussian.

**Note:** In their paper, Miller et al. derived a control variate for the case where $q_w$ is a fully-factorized Gaussian parameterized by its mean $\mu = [\mu_1, \ldots, \mu_d]$ and standard deviation $\sigma = [\sigma_1, \ldots, \sigma_d]$ ($w = \{\mu, \sigma\}$). That is, $q_w(z) = \mathcal{N}(z|\mu, \operatorname{diag}(\sigma^2))$. However, in their code (publicly available) they use a different parameterization. Instead of using $\sigma$, they use a different set of parameters, $\psi$, to represent the log of the standard deviation of $q_w$. That is, $q_w(z) = \mathcal{N}(z|\mu, \operatorname{diag}(e^{2\psi}))$. In order to explain, replicate and compare against the method they use, we derive the details of their approach for the latter case. (This derivation is not present in their paper, but follows all the steps closely.)

Miller et al. introduced a control variate to reduce the variance of the estimator of the gradient with respect to the mean parameters $\mu$ and a control variate to reduce the variance of the estimator of the gradient with respect to the log-scale parameters $\psi$. We will denote these control variates $c_\mu(w, \epsilon)$ and $c_\psi(w, \epsilon)$, respectively. Their main idea is to use curvature information about the model (via its Hessian) to construct both control variates. The control variate they propose for the mean parameters $c_\mu(w, \epsilon)$ can be computed efficiently via Hessian-vector products. On the other hand, the original proposal for $c_\psi(w, \epsilon)$ requires computing the (often) intractable Hessian $\nabla^2 f(\mu)$. To avoid this the authors propose an alternative control variate $\tilde{c}_\psi(w, \epsilon)$ based on some tractable approximations.

The authors noted that the use of these approximations lead to a significant deterioration of the control variate's variance reduction capability. However, no formal analysis that explained this was presented. We study these approximations in detail and explain exactly why this quality reduction is observed. Simply put, we observe that these approximations lead to a control variate that does not use curvature information about the model at all.

The rest of this section is organized as follows. In E.1, we present the resulting control variates obtained after applying the required approximations to deal with the intractable Hessian: $c_\mu(w, \epsilon)$ and $\tilde{c}_\psi(w, \epsilon)$. In E.2, we present Miller et al. original (intractable) control variate, $c_\psi(w, \epsilon)$, explain the source of intractability, and explain how the approximation used leads to the "weaker" control variate $\tilde{c}_\mu(w, \epsilon)$ presented in E.1. Finally, in E.3 we describe the drawbacks of the approach, and extend the approach to the case where $q_w$ is a Gaussian with a full-rank or diagonal plus low rank covariance matrix.

## E.1  Final control variate after approximations

Let $q_w(z)$ be the variational distribution. The gradient that must be estimated is given by

$$\nabla_w \mathop{\mathbb{E}}_{q_w(z)} f(z) = \nabla_w \mathop{\mathbb{E}}_{q_0(\epsilon)} f(\mathcal{T}_w(\epsilon)) \tag{17}$$

$$= \mathop{\mathbb{E}}_{q_0(\epsilon)} \nabla_w f(\mathcal{T}_w(\epsilon)) \tag{18}$$

$$= \mathop{\mathbb{E}}_{q_0(\epsilon)} \left( \frac{d\,\mathcal{T}_w(\epsilon)}{d\,w} \right)^\top \nabla f(\mathcal{T}_w(\epsilon)), \tag{19}$$

where $\nabla f(\mathcal{T}_w(\epsilon))$ is $\nabla f(z)$ evaluated at $z = \mathcal{T}_w(\epsilon)$. The gradient estimator obtained with a sample $\epsilon \sim q_0$ is given by

$$g(\epsilon) = \left( \frac{d\,\mathcal{T}_w(\epsilon)}{d\,w} \right)^\top \nabla f(\mathcal{T}_w(\epsilon)). \tag{20}$$

Miller et al. [17] propose to build a control variate using an approximation $\nabla \hat{f}(z)$ of $\nabla f(z)$. The control variate is given by the difference between the gradient estimator using this approximation and its expectation,

$$c(w, \epsilon) = \left(\frac{d\,\mathcal{T}_w(\epsilon)}{d\,w}\right)^\top \nabla \hat{f}(\mathcal{T}_w(\epsilon)) - \mathop{\mathbb{E}}_{q_0(\epsilon)} \left(\frac{d\,\mathcal{T}_w(\epsilon)}{d\,w}\right)^\top \nabla \hat{f}(\mathcal{T}_w(\epsilon)). \tag{21}$$

The quality of the control variate directly depends on the quality of the approximation $\nabla \hat{f}$. If $\nabla \hat{f}$ is very close to $\nabla f$, the control variate is able to approximate and cancel the estimator's noise. On the other hand, bad approximations lead to a small (or none) reduction in variance.

This idea is applied to fully-factorized Gaussian with parameters $\psi$ representing the log-scale. The reparameterization transformation is given by

$$\mathcal{T}_w(\epsilon) = \mu + e^\psi \odot \epsilon, \tag{22}$$

where $\odot$ is the element-wise product between vectors. The parameters are $w = (\mu, \psi)$. The control variate is derived differently for $\mu$ and $\psi$. We discuss the two cases separately.

**Control variate for $\mu$.** For $\mu$, the authors set $\nabla \hat{f}(z)$ to be a first order Taylor expansion of the true gradient around $\mu$. That is, $\nabla \hat{f}(z) = \nabla f(\mu) + \nabla^2 f(\mu)(z - \mu)$, where $\nabla^2 f(\mu)$ is the Hessian of $f$ evaluated at $z = \mu$. Then, it is not hard to show that the control variate becomes[3]

$$c_\mu(w, \epsilon) = \nabla^2 f(\mu)(e^\psi \odot \epsilon). \tag{27}$$

This control variate can be computed efficiently using Hessian-vector products, and will be effective when the approximation $\nabla \hat{f}(z)$ is close to $\nabla f(z)$ for $z \sim q_w(z)$.

The following derivation for $\psi$ is different from that given by Miller et al. We show that it is equivalent in Sec. E.2.

**Control variate for $\psi$.** For $\psi$, it is necessary – in order to obtain a closed-form expectation – to use a *constant* approximation of the form $\nabla \hat{f}(z) = \nabla f(\mu)$ (using the first order Taylor expansion as for $c_\mu(w, \epsilon)$ leads to intractable terms, see Section E.2). Then, it turns out that the expectation part of the control variate is zero, and so the control variate becomes[4]

$$\nabla \hat{f}(\mathcal{T}_w(\epsilon)) = \nabla f(\mu) + \nabla^2 f(\mu)(\mathcal{T}_w(\epsilon) - \mu) = \nabla f(\mu) + \nabla^2 f(\mu)(e^\psi \odot \epsilon). \tag{23}$$

The Jacobian of $\mathcal{T}$ with respect to $\mu$ is $\frac{d\,\mathcal{T}_w(\epsilon)}{d\,\mu} = I$. Then, we can calculate that

$$c_\mu(w, \epsilon) = \left(\frac{d\,\mathcal{T}_w(\epsilon)}{d\,\mu}\right)^\top \nabla \hat{f}(\mathcal{T}_w(\epsilon)) - \mathop{\mathbb{E}}_{q_0(\epsilon)} \left(\frac{d\,\mathcal{T}_w(\epsilon)}{d\,\mu}\right)^\top \nabla \hat{f}(\mathcal{T}_w(\epsilon)) \tag{24}$$

$$= \nabla f(\mu) + \nabla^2 f(\mu)(e^\psi \odot \epsilon) - \mathbb{E}\left[\nabla f(\mu) + \nabla^2 f(\mu)(e^\psi \odot \epsilon)\right] \tag{25}$$

$$= \nabla^2 f(\mu)(e^\psi \odot \epsilon). \tag{26}$$

[4]In this case the Jacobian of $\mathcal{T}$ with respect to $\psi$ is $\frac{d\,\mathcal{T}_w(\epsilon)}{d\,\psi} = \mathrm{diag}(e^\psi \odot \epsilon)$. It follows that

$$\tilde{c}_\psi(w, \epsilon) = \left(\frac{d\,\mathcal{T}_w(\epsilon)}{d\,\psi}\right)^\top \nabla \hat{f}(\mathcal{T}_w(\epsilon)) - \mathop{\mathbb{E}}_{q_0(\epsilon)} \left(\frac{d\,\mathcal{T}_w(\epsilon)}{d\,\psi}\right)^\top \nabla \hat{f}(\mathcal{T}_w(\epsilon)) \tag{28}$$

$$= \mathrm{diag}(e^\psi \odot \epsilon)\nabla f(\mu) - \mathop{\mathbb{E}}_{q_0(\epsilon)} \mathrm{diag}(e^\psi \odot \epsilon)\nabla f(\mu) \tag{29}$$

$$= e^\psi \odot \epsilon \odot \nabla f(\mu) - \mathop{\mathbb{E}}_{q_0(\epsilon)} e^\psi \odot \epsilon \odot \nabla f(\mu) \tag{30}$$

$$= e^\psi \odot \epsilon \odot \nabla f(\mu) \tag{31}$$

$$\tilde{c}_\psi(w, \epsilon) = e^\psi \odot \epsilon \odot \nabla f(\mu) \tag{32}$$

It can be observed that $\tilde{c}_\psi(w, \epsilon)$ does not use curvature information about the model. This control variate will be effective only in cases where $\nabla f(\mu)$ is close to $\nabla f(z)$ for $z \sim q_w(z)$.

## E.2 Original Derivation

Miller et al. [17] gave a more elaborate derivation of the above control variate for $\psi$. They start with the same first-order Taylor expansion $\nabla \hat{f}(z) = \nabla f(\mu) + \nabla^2 f(\mu)(z - \mu)$ as used for $\mu$. Applied directly, this suggests the control variate[5]

$$c_\psi(w, \epsilon) = \left( \nabla f(\mu) + \nabla^2 f(\mu)(e^\psi \odot \epsilon) \right) \odot \epsilon \odot e^\psi - \underbrace{\mathbb{E}_{q_0(\epsilon)} \left( \nabla f(\mu) + \nabla^2 f(\mu)(e^\psi \odot \epsilon) \right) \odot \epsilon \odot e^\psi}_{\mathrm{diag}(\nabla^2 f(\mu)) \odot e^{2\psi}}. \tag{37}$$

The first term from Eq. 37 can be computed efficiently using Hessian-vector products. The second term, however, is often intractable, since it requires the diagonal of the Hessian. In such cases, the authors propose to apply a further estimation process to estimate it using a baseline [1, 19]. The idea is that often gradients are estimated in a minibatch, based on a set of samples $\epsilon_1, \ldots, \epsilon_N$. Then, the expectation can be estimated without bias using the other samples in the minibatch. This results in the control variate for sample $i$ of

$$c_\psi(w, \epsilon_i) = \left( \nabla f(\mu) + \nabla^2 f(\mu)(e^\psi \odot \epsilon_i) \right) \odot \epsilon \odot e^\psi - \underbrace{\frac{1}{N-1} \sum_{\substack{j=1 \\ j \neq i}}^N \left( \nabla^2 f(\mu)(e^\psi \odot \epsilon_j) \right) \odot \epsilon_j \odot e^\psi}_{\text{baseline}}. \tag{38}$$

At a first glance it may appear that this control variate uses curvature information from the model via the Hessian $\nabla^2 f(\mu)$. However, a careful inspection shows that all these terms cancel out. The control variate for the full minibatch is simply

$$c_\psi(w, \epsilon_1, \cdots, \epsilon_N) = \sum_{i=1}^N c_\psi(w, \epsilon_i) = \sum_{i=1}^N \nabla f(\mu) \odot \epsilon_i \odot e^\psi. \tag{39}$$

This, of course, is exactly the same as taking a minibatch of the control variate derived in Eq. 32. Thus, the ideas of minibatch and baseline may somewhat obscure what is happening. It is not necessary to invoke the machinery of a baseline, nor to draw samples in a minibatch. A zero-th order Taylor expansion is equivalent, and has the practical advantage of remaining valid with a single sample. While some details of the baseline procedure were not available in the published paper, we confirmed this is equivalent to the control variate used in the publicly available code.

$$c_\psi(w, \epsilon) = \left( \frac{d\mathcal{T}_w(\epsilon)}{d\psi} \right)^\top \nabla \hat{f}(\mathcal{T}_w(\epsilon)) - \mathbb{E}_{q_0(\epsilon)} \left( \frac{d\mathcal{T}_w(\epsilon)}{d\psi} \right)^\top \nabla \hat{f}(\mathcal{T}_w(\epsilon)) \tag{33}$$

$$= \mathrm{diag}(e^\psi \odot \epsilon) \left( \nabla f(\mu) + \nabla^2 f(\mu)(e^\psi \odot \epsilon) \right) - \mathbb{E}_{q_0(\epsilon)} \mathrm{diag}(e^\psi \odot \epsilon) \left( \nabla f(\mu) + \nabla^2 f(\mu)(e^\psi \odot \epsilon) \right) \tag{34}$$

$$= \left( \nabla f(\mu) + \nabla^2 f(\mu)(e^\psi \odot \epsilon) \right) \odot e^\psi \odot \epsilon - \mathbb{E}_{q_0(\epsilon)} \left( \nabla^2 f(\mu)(e^\psi \odot \epsilon) \right) \odot (e^\psi \odot \epsilon) \tag{35}$$

$$\tag{36}$$

Finally, we can observe that $\mathbb{E} \left[ \left( \nabla f(\mu) + \nabla^2 f(\mu)(e^\psi \odot \epsilon) \right) \odot \epsilon \odot e^s \right] = \mathrm{diag}(\nabla^2 f(\mu)) \odot e^{2\psi}$.

### E.3 Limitations of the approach and extensions

One limitation of the above approach is that the control variate for $\psi$ is not very effective. Unless the diagonal of the Hessian is tractable, it uses a very crude approximation for $\nabla f(z)$. Thus, one would naturally expect this control variate to perform worse when the diagonal of the Hessian is not tractable. Indeed, this can be observed in the results obtained by Miller et al. [17]. Table 1 in their paper shows that the tractable control variate (Eq. 32, tractable), leads to a variance reduction several orders of magnitude worse than the one obtained using the control variate based on the true Hessian (Eq. 37, often intractable to compute).

In their simulations, this relatively poor performance for $\psi$ does not represent a big inconvenience. That is because of the following empirical observation: when using a fully-factorized Gaussian as variational distribution most of the gradient variance comes from mean parameters $\mu$, where a much better approximation of $\nabla f$ can be used. However, our results in this paper show that with non fully-factorized distributions most of the variance is often contributed by the scale parameters (see Fig. 1).

A second limitation is that their approach requires manual distribution-specific derivations. More specifically, in order to use the control variate with another distribution the expectation

$$\mathbb{E} \frac{d\,\mathcal{T}_w(\epsilon)}{d\,w}^{\top} \nabla \hat{f}(\mathcal{T}_w(\epsilon))$$

must be computed. In order to do so, a closed form expression for the Jacobian of $\mathcal{T}_w(\epsilon)$ is required. (One cannot use automatic differentiation for this since a mathematical expression for the Jacobian is needed in order to derive the expectation). Thus, extending the approach to other variational distributions is not trivial, and the difficulty depends on the variational distribution chosen. We now present three cases, two for which the extension can be done without much work (full-rank and diagonal plus low rank Gaussians), and other for which the extension requires extensive calculations (Householder flows [32]).

**Full-rank Gaussian:** In this case we have $q_w(z) = \mathcal{N}(z|\mu, \Sigma)$. The parameters are $w = (\mu, S)$, where S parameterizes the covariance matrix as $SS^{\top} = \Sigma$, and reparameterization is given by $z = \mu + S\epsilon$. If we let $\text{vec}(S)$ be a vector that contains all rows of $S$ in order, we get that the required Jacobians are given by

$$\frac{d\,\mathcal{T}_w(\epsilon)}{d\,\mu} = I \quad \text{and} \quad \frac{d\,\mathcal{T}_w(\epsilon)}{d\,\text{vec}(S)} = \begin{bmatrix} \epsilon^{\top} & 0_d^{\top} & \dots & 0_d^{\top} \\ 0_d^{\top} & \epsilon^{\top} & \dots & 0_d^{\top} \\ & & \dots & \\ 0_d^{\top} & 0_d^{\top} & \dots & \epsilon^{\top} \end{bmatrix}, \tag{40}$$

where $\epsilon^{\top}$ is a row vector of dimension $d$ and $0_d^{\top}$ is the zero row vector of dimension $d$. The Jacobian $\frac{d\,\mathcal{T}_w(\epsilon)}{d\,\text{vec}(S)}$ has dimension $d \times d^2$. Following section E.1 and using the above expressions for the Jacobians we get

$$c_\mu(w, \epsilon) = \nabla^2 f(\mu) S\epsilon \quad \text{and} \quad \tilde{c}_S(w, \epsilon) = \nabla f(\mu)\epsilon^{\top}. \tag{41}$$

Both $c_\mu(w, \epsilon)$ and $\tilde{c}_S(w, \epsilon)$ can be computed efficiently.

**Diagonal plus low rank Gaussian:** In this case we have $q_w(z) = \mathcal{N}(z|\mu, \Sigma)$. The parameters are $w = (\mu, \psi, U)$, where $\mu$ and $\psi$ are vectors of dimension $d$, and $U$ is a matrix of size $d \times r$. The covariance is parameterized as $\Sigma = \text{diag}(e^{2\psi}) + UU^{\top}$. Reparameterization is given by $z = \mu + e^{\psi} \odot \epsilon_d + U\epsilon_r$, where $\epsilon_d$ and $\epsilon_r$ are independent samples of standard Normal distributions of dimension $d$ and $r$, respectively. In this case the required Jacobians are given by

$$\frac{d\,\mathcal{T}_w(\epsilon_d, \epsilon_r)}{d\,\mu} = I \,, \quad \frac{d\,\mathcal{T}_w(\epsilon_d, \epsilon_r)}{d\,\psi} = \text{diag}(e^{\psi} \odot \epsilon_d) \quad \text{and} \quad \frac{d\,\mathcal{T}_w(\epsilon_d, \epsilon_r)}{d\,\text{vec}(U)} = \begin{bmatrix} \epsilon_r^{\top} & 0_r^{\top} & \dots & 0_r^{\top} \\ 0_r^{\top} & \epsilon_r^{\top} & \dots & 0_r^{\top} \\ & & \dots & \\ 0_r^{\top} & 0_r^{\top} & \dots & \epsilon_r^{\top} \end{bmatrix}. \tag{42}$$

Following section [E.1] and using the above expressions for the Jacobians we get

$$c_\mu(w, \epsilon_d, \epsilon_r) = \nabla^2 f(\mu)(e^\psi \odot \epsilon_d + U\epsilon_r) \tag{43}$$

$$\tilde{c}_\psi(w, \epsilon_d, \epsilon_r) = \nabla f(\mu) \odot e^\psi \odot \epsilon_d \tag{44}$$

$$\tilde{c}_U(w, \epsilon_d, \epsilon_r) = \nabla f(\mu)\epsilon_r^\top. \tag{45}$$

**Householder flows:** In this case we have a Gaussian distribution with reparameterization given by $z = \mu + \prod_{i=1}^M H(v_i)D\epsilon$, where $M$ is the number of flow steps used, $D = \mathrm{diag}(\sigma)$ is a diagonal matrix, and $H_i$ is a matrix parameterized by vector $v_i$ as $H_i(v_i) = \left(I - 2\frac{v_i v_i^\top}{\|v_i\|^2}\right)$. The parameter set is given by $w = \{\mu, \sigma, v_1, \ldots, v_M\}$. In this case, computing the Jacobians required to apply Miller et al. approach is quite complex, because of the complex dependency of $\mathcal{T}_w$ on the parameters $v_i$.

## Footnotes

[3]To see this, observe that

[5]Again, $\frac{d\mathcal{T}_w(\epsilon)}{d\psi} = \mathrm{diag}(e^\psi \odot \epsilon)$ and $\nabla \hat{f}(\mathcal{T}_w(\epsilon)) = \nabla f(\mu) + \nabla^2 f(\mu)(e^\psi \odot \epsilon)$. We thus have that