[Reviews · NeurIPS 2020]

Review 1

Summary and Contributions: [Update] I've carefully read the authors' rebuttal and the other reviews. I am glad the authors included a figure in the rebuttal which reports ELBO curves wrt wall clock time. This makes it much clearer that the method converges substantially more quickly. (While in principle I can do such "visual scaling" in my head, in practice it's not so easy.) Consequently I am raising my score to a 6. That being said I think there are still a number of ways the authors could improve the paper in revision: - i think it's important to report actual variance reduction for the trio of experiments (not the ideal case reported in Figure 1) - add ELBO curves wrt wall clock time (as in the rebuttal) - in the same spirit i would argue it makes more sense for Figure 3 to use a fixed wall clock time instead of a fixed number of iterations - i find it strange that the authors' rebuttal examined downstream metrics for a new task with simulated data instead of one of the tasks in the paper. granted, there are issues with with evaluating wrt metrics like test error (especially for small datasets), but i would still argue that it would be useful to ground 3 nat differences, which are hard to interpret, in something that is more concrete. ----------------------------------------------- The authors describe an algorithm in which a learned quadratic control variate is used to reduce gradient variance in variational inference. This algorithm is reasonably widely applicable (at least for reparameterizable variational distributions with closed form means/variances) and can substantially reduce gradient variance.

Strengths: The method is relatively straightforward to implement and of fairly wide applicability. The additional computational cost required is also moderate.

Weaknesses: While the authors have convinced me that their method can lower gradient variance, they haven't convinced me that the resulting algorithm actually delivers what we ~actually~ care about, which is learning variational distributions that are meaningfully better (or at least doing the learning faster). To be concrete, the number of data points used in the Logistic Regression and BNN experiments is 700 and 100, respectively. This means that a normalized ELBO would be divided by 700 and 100, respectively. On the other hand, the (unnormalized) ELBO differences reported on these two models are of order ~3-4 nats. When these differences are normalized appropriately, the result is very small fractions of a nat per data point. It is extremely likely that if we were to evaluate on downstream metrics like predictive RMSE or predictive log likelihood, differences between the learned approximate posteriors would be essentially non-existent. This is not the author's fault, in the sense that this kind of negative result plagues a lot of the research in this area. The reality is that moderate gains in gradient variance don't necessarily buy you very much, i.e. they don't necessarily help you solve the optimization problem better. In order to convince me that this methodology is actually useful I would need to see something like: i) substantially faster convergence wrt wall clock time; ii) better robustness to bad initializations; iii) better variational optima as measured by some downstream task; etc. Unfortunately, it is not enough to demonstrate lower gradient variance, especially since there are already a number of methods that do that.

Correctness: I have no particular concerns about correctness and/or empirical methodology.

Clarity: The paper is generally well-written. Some points that I think could be better clarified include: - on line 141 the authors state "However, the second method introduces a smaller overhead." it would be helpful if this point were explained in greater detail. - on line 109ff: "We consider the case where Bv and Σw are parameterized as diagonal plus..." The author's method is applicable beyond the gaussian case, so it would be good if they make sure their discussion of computational complexity etc. is careful about when it might be assuming a gaussian variational distribution and when it's not.

Relation to Prior Work: There is quite a bit of work in this area so it would be good if the authors made additional effort to ensure they're not missing any prior work. Off the top of my head, two references that should probably be referenced and/or discussed include: [1] “Amortized variance reduction for doubly stochastic objectives,” Ayman Boustati, Sattar Vakili, James Hensman, ST John. [2] “Pathwise Derivatives for Multivariate Distributions,” Martin Jankowiak, Theofanis Karaletsos.

Reproducibility: Yes

Additional Feedback: - i think reporting normalized ELBOs would make it easier to interpret your results. - it would be valuable to include ELBO curves wrt wall clock time. - you report gradient variances in the "ideal case" (Figure 1) but not in the pragmatic case where the parameters v are not trained to completion. it would be very informative to report gradient variance as measured in your actual experiments (e.g. at the beginning of training, at iteration X, and at convergence). - i suggest that the authors could make their paper much more compelling if they could demonstrate situations in which their control variate yields more than small improvements in final ELBOs.


Review 2

Summary and Contributions: The authors proposed a new control variate method to reduce the gradient variance in stochastic gradient variational inference with a reparameterized gradient. In this paper, firstly, they considered approximating the joint distribution by a parameterized quadratic function \hat{f} and formulated the control variate by using this. Secondly, they used a "double descent" scheme to fit the parameters of \hat{f}. They also showed that, if the variational distribution is in a location-scale family, the deterministic term in the proposed control variate is tractable (computed in a closed-form), and therefore it is easy to compute. They empirically showed that the proposed control variate could reduce the variance of scale parameters and achieved a better optimization performance than that of Taylor-based control variates.

Strengths: ・The idea itself is somewhat novel. ・It seems that the proposed method is easy to implement and to be combined with the modern auto-differential toolbox such as Pytorch or Stan. ・The proposed method has the possibility to improve the optimization performance of SGVI (as I mentioned below, it is necessary to compare with more related work to show this.).

Weaknesses: There are mainly two points of weaknesses (limitations) in this paper; (a) the lack of theoretical guarantees for gradient variance or convergence, and (b) the weakness of experiments. (a) : lackness of theoretical guarantees In this paper, only the traceability of the proposed control variate is guaranteed (lemma3.1). It seems good, but in the variance reduction field, it is more important to analyze the behavior of gradient variance itself. The lack of this perspective occurs the problem that we can not understand when this method works well or not. For control variate, it is difficult to analyze this perspective. Therefore, this is the limitation of control variate research itself. (b) : Experiments I think this is the weakest point of this paper. The reasons are as follows. (b-1) : Baseline method In this paper, the authors compared the proposed method with the naive reparameterized gradient and the Taylor-based approach proposed by (Geffner and Domke, 2018). However, it is not enough to show the impact of the proposed method. There are three approaches to reduce the gradient variance in SGVI; control variate, importance-weighted method (e.g., Ruiz et al, 2017), and sampling-based method (e.g., Bachholz et al, 2018). If the performance of the proposed method is no more than that of these, the contribution becomes trivial. (As I mentioned below, it is necessary to compare with the other standard variance reduction methods.) (b-2) : Results In this paper, the authors compared the optimization performance per iteration. However, in the real world, the convergence speed is more important (and this is the main point the variance reduction research focuses on). It is uncertain that the proposed method truly improves the optimization convergence in this point of view. (I concern that the proposed method needs more time to complete the optimization procedure per iteration due to double descent.) Related work also reported the results of optimization not per iteration but per real-world time [e.g. (Ruiz et al., 2017) (Miller et al., 2018 ) (Buchholz et al, 2018)]. To confirm the optimization performance, the authors should show the experimental results of training ELBO on the basis of the wall-time clock. (b-3) : Settings of learning-rate for v Because of the double descent scheme, the performance depends on the initial learning rate obviously. However, in this paper, the authors only use $\alpha^{v}$ = 0.01. To understand the characteristics of the proposed method, it would be nice to show the experimental results for various $\alpha^{v}$ values.

Correctness: The theoretical claims seem to be correct, but there are some possibilities to be incorrect in the other claims. For example, in experimental results, the author said "our method improves over competing approaches". However, I can not confirm that the proposed method truly improves the optimization convergence because they compared the performance based on the iteration.

Clarity: While the paper is pretty readable, there is certainly room for improvement in the clarity of the paper. (a) : Related work This paper is missing a related work section. The variance reduction for SGVI is a hot topic in ML fields, and therefore there are many studies on it, e.g.; ・(Xu et al., 2018); theoretically investigated the variance reduction properties on reparameterized gradient ・(Bachholz et al., 2018); introduced the sophisticated MC method (RQMC) into SGVI with theoretical guarantee ・(Domke, 2019); gave bounds for the common “reparameterization” estimators and showed he's assumption is the best possible assumption because these bounds are unimprobable. Even if they only focus on the improvement of the control variates method, it is preferable to summarize these related work as the above and discuss the relationship between them to insist on the novelty of the proposed method. Furthermore, these studies may help you to analyze the gradient variance of the proposed method. (b) : trivial Probably, they should remove the section number for the broader impact section. This part is defined as an additional statement like Reference or Acknowledgements.

Relation to Prior Work: For control variate, It seems to be enough. However, to insist on the contribution and novelty of this work, the author should make a "related work" section and summarize more methods in variance reduction such as the RQMC-based method. Furthermore, they should compare with more previous methods to show the usefulness of the proposed method.

Reproducibility: Yes

Additional Feedback: Questions : (1). In Method 1. and 2., they introduced the unbiased estimator of CV gradient. However, I think this CV gradient itself has the estimation variance due to MC estimation. Could this variance cause the negative effects for the update of v? If no, why? Comments : ・Performance on the test dataset In this paper, the author reported the many experimental results for the training dataset (e.g. training ELBO). It is really good, but it would be nice if they report the results on the test dataset to show the predictive performance. I recommend using predictive log-likelihood to show this. If the performance of the proposed method is better in real-time as a result of comparative experiments between more methods and the performance of the optimization in real-time, this paper will turn out to be a strong one. ========================= After reading the author response ========================= I read all of the reviews and the authors' rebuttal. My concerns have been allayed. If the author(s) would reflect all of the contents in their feedback, I think this paper's novelty and impact are enough to accept by NeurIPS. Therefore, I decided to increase my score from 4 to 7. After reading this paper again, I think this paper becomes more interesting and easier to understand the effective of the proposed approach if the author(s) report the experimental results of how the gradient variance changes as optimization proceeds.


Review 3

Summary and Contributions: ...Update... I increased my score to 8. My initial assessment was at 5-7. The authors adequately addressed my question about how the rank affects the performance, so I increased to 7. But I decided to increase by 1 more point, because I think my initial reluctance to suggest the paper may have been due to my bias---I just didn't find the ideas in the paper very interesting; mostly the key ideas exist in prior literature, but are combined together in a sensible way. However, I think the paper is solid, and the method is simple, practical, has a wide applicability, and is convincingly shown to lead to improved performance. I could see the method being widely used to reduce reparameterization gradient variance. The idea to use a quadratic approximation is a fundamental one (it's a linear approximation of the gradient, hence the simplest possible approximation scheme), so it may deserve another examination as is done in this paper. I added a few more comments below for the authors to aid in improving clarity in some places (marked with *Update*). I also think the paper could be further improved by showing experiments of how the gradient variance changes as learning progresses. But it's not such a key point; the experiments are already convincing to show the usefulness of the method. -------------------------------------------- Previously Miller et al. created a method for reducing reparameterization gradient variance using a Taylor series quadratic expansion of the model to use as a control variate. However, that method was only applied to factorized Gaussian distributions, and there are difficulties with applying the method to distributions with full covariance structure, because it would require computing the full Hessian of the model, which is computationally heavy. The current paper aims to tackle this issue by instead of performing a Taylor series expansion, keeping a running parameterized quadratic approximation of the model to use as a control variate. The quadratic approximation can be represented using a diagonal + low-rank structure, which gives reasonable theoretical scaling to larger problems. The quadratic approximation is learned online using a double-descent scheme, similar to several prior works. The proposed method gives 2-3 orders of magnitude reduction in gradient variance for the scale parameters in addition to the mean parameters of the distribution, while giving only a modest overhead in computational time. Beyond proposing the method, the papers contributions include the experimental analysis, which implies that the previous method by Miller does not help much for reducing the gradient variance for the scale parameters.

Strengths: Previous control variates using quadratic functions are not practical for non-factorized distributions. The paper explained the flaws of the previous method both theoretically and experimentally. And they provided a solution, which gives significant reduction in gradient variance, and is also practical. The experiment in Figure 1 showed that the gradient variance can be dominated by the scale parameters, which shows the flaws of the previous method, and provides a good motivation for a new method. The experiments in the previous Miller paper also showed that the variance reduction for scale paramaters for that method can be low giving further evidence that the claim is correct (in the previous work the gradient variance was dominated by the mean parameters, so the method worked okay in that setting, but in the non-factorized setting considered in this paper, the scale parameters are more important, so Miller's method does not work well). The rank of the quadratic approximations was r = 10 for d=(120, 37) dimensional problems, and r = 20 for a d=653 dimensional problem, which implies that the rank of the approximation can be much lower than the dimensionality of the problem, and hence the method might scale favorably. It was also good that both computational time as well as iteration count aspects of the method were explored.

Weaknesses: I found the novelty of the work low. The idea to use a quadratic was first done by Miller. The double-descent scheme is used in several prior works. Method 2, which tries matching the gradients is also known, e.g. see "Sobolev training for neural networks". Some ablation studies are missing. My biggest concern was that they did not include experiments examining what the effect of the rank r of the quadratic approximation is on the performance. It would have been good to see experiments with only a diagonal quadratic approximation, r=5, r=40, etc. For example, one of the main claims of the paper was that the scale parameters dominate the gradient variance in the full covariance structure Gaussian case. One might then expect that using only a diagonal quadratic may not give much reduction in gradient variance. It would be good to see such experimental analysis. Such experiments would also be good to indicate how large of a rank is necessary in practice to achieve gradient variance reductions; this will determine the expected computational cost.

Correctness: Mostly yes, as far as I could tell. I was not able to verify the derivations in Appendix D, as it appears to be explained differently to Miller's work, and also because Miller's work did not explain their method fully. I think the claim in D.3 about the prior method requiring distribution specific derivations being a limitation compared to your method is not quite accurate. The claim was that Miller's method requires a formula for the reparameterized gradient, and computing the expectation analytically (which can not be done with automatic differentiation), while the new method only needs the mean and variance of the distribution, and can use automatic differentiation. First, the switch to using automatic differentiation for the expectation gradient could also be done with the Taylor expansion method, if one just plugs in the Taylor expansion quadratic parameters into your method. Secondly, your method also requires manual derivations of the mean and variance of the distribution, and it may not be clear how to immediately do this. For example, you make the example with Householder flows where you say your method would be better, but it's not immediately clear to me what the mean and variance of this distribution would be. I tried calling mean() on the pyro implementation of Householder flows, but it gave a Not Implemented Error. *Update* I took another look, and saw that it's obvious, but maybe you can make it even more obvious by just writing out "The mean is... the variance is ..."

Clarity: The main bits of the paper, including the motivation and description of the method were well explained. I found some of the more subtle points in appendices D.1 and D.2 confusingly written. In particular Eq. 24 is not what Miller et al. claim to do. They claim to estimate the curvature H as hat{H} using an unbiased estimator due to Bekas, then use this curvature to compute the expectation. This is not aligned with Eq. 24, which does not include an explicit estimator for the Hessian, but instead has a "baseline" term. If you confirmed that your claim is true, that's OK. But the explanation at the moment appears incomplete. In particular explaining how this fits together with the Hessian approximator from Bekas would complete the picture. *Update* I believe that the authors correctly checked this, as they compared with the code. I took another look, and was able to verify the derivation. For the first part, I think it may help to write out H=0, with an equation, and remind the reader inside the lower equation why it's dropped. For the D.2 derivation, I understood the derivation, but what I don't understand is why there is a Hessian term? In Miller's method the Hessian is never explicitly computed, so it seems that you are saying that Miller's method is implicitly doing that computation. It would be good to explain where that Hessian comes from. Other: On line 87, I believe Eq 37 should be Eq 23 (some other equation numbers are also messed up, which seems to be caused by the Equation numbers restarting from 1 in the appendix). In Eq 24, the epsilon on the left should be epislon_i. "However, a careful inspection shows that all these terms cancel out." It may be better to write out that the summation across (N-1) cases cancels with the bit on the left side.

Relation to Prior Work: The paper could have done this better in some instances. For example, the expectation of a quadratic given in Lemma 3.1. is well-known, e.g. https://en.wikipedia.org/wiki/Quadratic_form_(statistics) But it is nowhere mentioned, that this is well-known. Also, the double-descent scheme is used in many prior works, but the citations are left until later. It would be important to cite prior works when you first introduce the double-descent idea to avoid readers thinking that you invented the method. Also, there's a prior work which considers variance reduction for non-factorized distributions. The method is different, but it shares some similarities,e g. using a double-descent scheme: Jankowiak and Karaletsos "Pathwise Derivatives for Multivariate Distributions" AISTATS 2019 It would be worth mentioning or even comparing to.

Reproducibility: Yes

Additional Feedback: My main concern about the paper was a lack of experiments exploring the effect of the rank of the quadratic approximation, so I would appreciate if the authors can address this in the rebuttal. I believe the score should be between 5-7, and in principle the method is sound and well motivated to tackle issues with non-factorized distributions. I gave a 5 for now, and may increase based on the rebuttal. Line 85: "We propose to find... double-descent scheme..." This is again misleading, as you did not provide a citation to prior work, whose double-descent scheme you are using. Line 42: citation is missing for the double descent procedure from prior work. Line 45: "leads orders" -> "leads to orders" Line 79: "it's" -> its Line 155: "specially" -> especially Line 195: "it is" -> is Line 201: "the the" -> the Please add the dimensionality, d, of the datasets also into the main paper. This is important to quickly understand and compare to the rank of the approximations. The order of the expectation and regular term in the definition of the control variate is inconsistent between the main paper and appendix D. In one case, the expectation is subtracted, while in the other this is flipped, and the expectation is added. In the appendix equation 15, it seems that there is a term missing in the expectation: there should be a H*sigma^2*epsilon^2 term, which will not be 0. I believe this is deliberate, but I didn't understand why it is omitted. Instead of using the method used by Miller to approximate the Hessian diagonal, it could be computed exactly using packages such as BackPack. But computing the diagonal would still be computationally expensive, so probably your method may still have an advantage. Jankowiak and Karaletsos "Pathwise Derivatives for Multivariate Distributions" AISTATS 2019 Felix Dangel, Frederik Kunstner, Philipp Hennig, "BackPACK: Packing more into backprop", ICLR 2020 Wojciech Marian Czarneckiet al,"Sobolev Training for Neural Networks", NeurIPS 2017.


Review 4

Summary and Contributions: The paper introduces a new control variate for reparameterisable variational families. It is based on a quadratic function whose expectation with respect to the variational distribution can be computed analytically using the first two moments of the variational distribution. The method seems to offer larger variance reductions compared to control variates based on a Taylor-approximation, particularly for low-rank Gaussian variational densities. ################################### POST AUTHOR-RESPONSE UPDATE Having read the response from the authors and the other reviews/discussion, I will keep my score of weak accept. Generally, I think the idea to use a quadratic function with tractable expectation is interesting and new in that form and seems to perform well. The main criticism in my review was that I felt that some additional experimental evaluation would be useful such as a of variational family different from a Gaussian or with a different covariance structure. Of course, not addressing these should not be a reason for rejection! The rebuttal also commented adequately on my point of criticism with respect to the Sticking the landing estimator. ###################################

Strengths: The new control variate is well motivated and is different from previously considered variance reduction techniques. The idea to use a quadratic function with tractable expectation is interesting. The control variates can be computed without much overhead. The approach can yield gradients with lower variance compared to a previously-considered control variate using a Taylor expansion, which is also demonstrated empirically.

Weaknesses: Whereas it has been demonstrated empirically that the proposed approach can improve on a Taylor-based control variate, it is not so clear how the control variate compares to other variance reduction techniques (such as for example Roeder et al., Sticking the Landing: Simple, Lower-Variance Gradient Estimators for Variational Inference). Also how do the results change if one optimizes the weight \gamma, or if one uses a variational family different from a Gaussian? Adding some experiments that address these questions would I think increase the relevance to the NeurIPS community.

Correctness: The claims seem correct with details provided in the appendix. The empirical methodology looks correct as well.

Clarity: The paper is well written and polished.

Relation to Prior Work: Related work is largely cited.

Reproducibility: Yes

Additional Feedback: How does it relate to the Taylor approximations used in ref. [22] Paisley et al, Variational Bayesian Inference with Stochastic Search, 2012? Is it possible to consider polynomials of higher order than two as control variates (assuming known higher moments for the variational distribution)? Is there also a relative large improvement over ref [16] if the Gaussian variational distribution comes from a linear flow (e.g. Householder flow) as in the low-rank case, or is there less variance? Why does Method 1 for optimizing v not perform better - does it have a higher variance itself?

[Author Response · NeurIPS 2020]

(R1: The distribution learned is not better for downstream tasks. Is ELBO the right metric? Are 3 nats relevant? ...)

**\* Why ELBO?** The ELBO is widely accepted and used in the VI community, and a lot of work compares methods in

terms of speed and convergence wrt ELBO (Miller et al. NIPS 2017, Buchholz et al. 2018, among others), showing

ELBO improvements and speedups similar to ours. Our work sits in this program.

**\* Why not test error?** The ELBO is the quantity that VI optimizes, it is more stable than test error (it does not depend

on validation data – see results below), and it does not depend on model miss-specification (test error does).

**\* Should use normalized ELBO.** We don't see why we should normalize the ELBO wrt the size of the dataset (and

no references were provided to support the use of normalized ELBOs). The ELBO measures improvements in KL

divergence; a measure between two distributions over the random variable $z$, not over the data.

**\* Are 3 nats significant?** There are cases where a divergence of 3 nats is certainly significant. Take some posterior

$p(z|x)$, and choose a set $A$ such that $p(z \in A|x) = 0.05$. Choose $q(z)$ to be $q(z) = 20p(z|x)$ for $z \in A$ and $q(z) = 0$

if $z \notin A$. In this case we get $\text{KL}(q(z)||p(z|x)) = \log 20 \approx 3$, despite the distributions being quite different.

**\* With a dataset of size 100 and a difference of 3-4 nats it is extremely likely that if we were to evaluate on**

**downstream metrics [...] differences between the learned approximate posteriors would be essentially non-**

**existent.** We think this claim is unfounded, and no references were provided. We ran simulations to support our position.

We consider a 10 dimensional logistic regression model with a dataset of size 100. We obtain approximating distributions

that achieve different ELBO values by running VI for different numbers of iterations. For each approximating distribution

we use a test set to measure classification test error. We repeat this 100 times. Results are shown next.

We see that a difference of 3 nats

leads to improvements of around

0.2% in test error. This contradicts

the claim that that the a 3 nats im-

provement leads to non-existent im-

provements in performance on down-

stream metrics.

(R1, R2: Convergence wrt wall clock time.) The paper shows the time cost of each method in Table 1. While results wrt

to wall clock time may be obtained from the table, we'd be happy to add them to the paper. For instance, the two leftmost

figures below show some examples (two models, diagonal plus low rank $q$, best step-size chosen retrospectively). Our

method achieves speedups of at least $3\times$. Results for other cases are similar, with speedups ranging from $3\times$ to $7\times$.

(R2, R4: Comparison to other methods.) We focused on the development of a control variate (CV) that addressed the

issues of the one proposed by Miller et al. (2017). Our CV may be used jointly with other variance reduction methods,

and thus should not be seen as a replacement but as an addition to them. For instance, the sticking-the-landing (STL)

estimator (Roeder et al. 2017) can be used with our CV in two ways: a) setting the base gradient estimator to the STL;

b) creating the "STL control variate" and using in concert with our CV (Geffner and Domke, 2018). Our CV can also

be used with Randomized QMC sampling (Buchholz et al 2018). Finally, while we focus on reparameterization, our

CV could be used with other estimators as well, such as the score function or generalized reparameterization (Ruiz et al.

2016), as long as the covariance of the variational distribution is known. This is done by obtaining the second term from

eq. 5 using the corresponding estimator (instead of reparameterization). We'll add a discussion about this in the paper.

(R3: Low novelty.) We respectfully disagree. Miller et al. used a quadratic function for the mean parameters, but a

linear function for the scale (problematic with non-factorized distributions). We provide an extensive new analysis of

this. Also, double-descent was previously used, but for the score function estimator with importance sampling (Ruiz et

al. 2016) or with relaxations for discrete variables (Tucker et al. 2016, Grathwohl et al. 2018). We use these ideas

jointly in a different way to address the main issue with the CV by Miller et al. (uncovered with our new analysis).

(R3: Rank use for the control variate.) We chose ranks 10 and 20 to show that even with ranks considerably smaller than

the dimension of the problem the control variate lead to improved results. Results for more ranks could be insightful,

and we'll happily add them to the paper. For instance, the two rightmost figures below show results for different ranks.

(R3: Description of method by Miller et al.) We gave a lot of thought to how to present this. We resolved the ambiguities

in their paper by looking at the code, which reflects the exact method used for their experiments. (We ran simulations

with the exact same models and random seeds to verify that the method we present and the one used in their code are

equivalent.) We believe they mention the method by Bekas because they use matrix-vector products to estimate the

trace of a matrix (the scaled Hessian). However, with the analysis we present, we show that the way they combine this

with baselines leads to the undesirable cancellations described in our paper.

53

[Meta-Review · NeurIPS 2020]

The paper introduces an improvement to the quadratic control variant of Miller et al. for reducing the reparameterization gradient variance. The key idea is to use a parameterized quadratic approximation to the model and learn the parameters using a double-descent scheme. This is shown to perform better than the Taylor-expansion based approach of Miller et al., with an important advantage of reducing the gradient variance of not only the mean but also the scale parameters. The reviewers had some concerns about the novelty and the empirical evaluation of the method but these have been convincingly addressed in the author response. The authors are urged to incorporate the suggestion made by the reviewers to improve the paper further.